# Angiopoietin 1 Attenuates Dysregulated Angiogenesis in the Gastrocnemius of DMD Mice

**DOI:** 10.3390/ijms252111824

**Published:** 2024-11-04

**Authors:** Andrew McClennan, Lisa Hoffman

**Affiliations:** 1Department of Medical Biophysics, Western University, London, ON N6A 5C1, Canada; andrew.mcclennan@cuanschutz.edu; 2Imaging Program, The Lawson Health Research Institute, London, ON N6C 2R5, Canada; 3Department of Anatomy and Cell Biology, Western University, London, ON N6A 5C1, Canada

**Keywords:** angiopoietin-1, Duchenne muscular dystrophy (DMD), inflammation, mdx/utrn+/−, microvasculature, vascular therapy

## Abstract

Duchenne muscular dystrophy (DMD) is a degenerative neuromuscular disease caused by a lack of functional dystrophin. Ang 1 paracrine signalling maintains the endothelial barrier of blood vessels, preventing plasma leakage. Chronic inflammation, a consequence of DMD, causes endothelial barrier dysfunction in skeletal muscle. We aim to elucidate changes in the DMD mouse’s gastrocnemius microvascular niche following local administration of Ang 1. Gastrocnemii were collected from eight-week-old mdx/utrn+/− and healthy mice. Additional DMD cohort received an intramuscular injection of Ang 1 to gastrocnemius and contralateral control. Gastrocnemii were collected for analysis after two weeks. Using immunohistochemistry and real-time quantitative reverse transcription, we demonstrated an abundance of endothelial cells in DMD mouse’s gastrocnemius, but morphology and gene expression were altered. Myofiber perimeters were shorter in DMD mice. Following Ang 1 treatment, fewer endothelial cells were present, and microvessels were more circular. *Vegfr1*, *Vegfr2*, and *Vegfa* expression in Ang 1-treated gastronemii increased, while myofiber size distribution was consistent with vehicle-only gastrocnemii. These results suggest robust angiogenesis in DMD mice, but essential genes were underexpressed—furthermore, exogenous Ang 1 attenuated angiogenesis. Consequentially, gene expression increased. The impact must be investigated further, as Ang 1 therapy may be pivotal in restoring the skeletal muscle microvascular niche.

## 1. Introduction

Duchenne muscular dystrophy (DMD) is a genetic disorder in which the protein dystrophin loses functionality and cannot distribute contractile forces, resulting in recurring sarcolemma microlesions. This damage to muscle fibres leads to muscle wasting and chronic inflammation. Inflammatory cytokines impair the microvascular niche’s functionality [1]. In 1968, Demos et al. characterized blood flow and capillary alterations in clinical DMD [2]. Contemporary evidence shows that endothelial cells of mdx mice, a standard animal model of DMD, have impaired function [3] and angiogenic capabilities [4] that lead to ischemia. Latroche et al. reported structural and functional microvascular niche alterations in mdx mice and stated the importance of devising new vascular-targeted therapies [5]. Rescuing the microvasculature of affected muscles may improve the efficacy of restorative therapies for patients with DMD.

Vascular-targeted therapies being developed aim to reduce ischemia and improve the microenvironment of chronically inflamed muscles. When treated with vascular endothelial growth factor (VEGF), a potent inducer of angiogenesis and myogenesis, mdx mice show improved muscle strength [6,7]. In 2021, Xin et al. showed improved efficacy of mini-dystrophin gene therapy when paired with VEGF in dystrophin/utrophin double knockout mice, another mouse model with more severe pathology than mdx mice [8]. VEGF is critical in the initial vascular plexus, but does not instigate endothelial cell maturation. Furthermore, VEGF overexpression increases vascularity, but disrupts vascular barrier function, causing vessels to leak [9]. VEGF-induced neovasculature may be abundant, but evidence of functional improvements is lacking. Neither intramuscular injection of VEGF nor VEGF gene therapy alone reduces profound fibrosis in murine models of DMD [8,10]. Exploration of regulator factors that contribute to vessel stability has yielded intriguing results.

The tyrosine kinase receptor, Tie2, shares responsibility for regulating the vascular endothelial barrier in quiescent endothelial cells. Angiopoietin 1 (Ang 1) and angiopoietin 2 (Ang 2) are the main ligands for Tie2 [11]. While quiescent, Ang 1 binds to Tie2. The ensuing signalling cascade results in vascular endothelial (VE) cadherin remaining localized to the endothelial cell surface. VE-cadherin increases adhesion between endothelial cells by forming homophilic, intercellular junctions, ensuring vessel stability and function [12,13]. Moreover, similar to VEGF, Ang 1-induced Tie2 activation regulates endothelial cell survival [14]. Angiopoietin 3 does not activate Tie2 receptors and inhibits Ang 1 activation. Angiopoietin 4 activates Tie2 receptors, but is only found in human lungs [15].

During the inflammatory response, proinflammatory cytokines VEGF and Ang 2 bind vascular endothelial growth factor receptor 2 and Tie2 receptors on endothelial cells, respectively. In conjunction, these promote VE-cadherin internalization, causing vascular permeability to increase [16,17]. This healthy response is a prerequisite for angiogenesis. Typically, as inflammation subsides, the endothelial cell barrier is restored, and endothelial cells return to a quiescent state. This process response becomes detrimental when inflammation is chronic, such as in cases of DMD. Excitingly, Thurston et al. report that vessels in Ang 1-overexpressing mice are resistant to leaks caused by inflammatory factors [9], positing Ang 1’s potential to reduce chronic inflammation-induced microvascular leakage.

Previous investigation into the perfusion changes in gastrocnemius muscles of mdx/utrn+/− mice using Dynamic Contrast-Enhanced Computed Tomography following intramuscular injection of Ang 1 shows a reduction in fibrosis and an increase in blood volume relative to healthy controls [10]. Recent evidence demonstrates the Ang 1/Ang 2 protein ratio is chronically low, which correlates with decreased Tie2 activation. This suggests dysregulation of Ang 1/Tie2 signalling in the diaphragm of mdx/utrn+/− mice [18]. Considering VEGF’s enhancement of mini-dystrophin gene therapy efficacy [8] and Ang 1’s anti-inflammatory properties, Ang 1 may prove invaluable as an adjunct vascular-therapeutic agent for treating DMD.

We aim to elucidate changes in microvessel architecture and gene expression following the intramuscular injection of Ang 1 into the gastrocnemius muscle of mdx/utrn+/− mice.

## 2. Results

Healthy and Diseased mice were sacrificed at 8 weeks of age to evaluate initial muscle architecture and gene expression. These results established that 8 weeks of age was an appropriate intervention time point. Mice that received treatment were sacrificed after two weeks of recovery. We identified changes in endothelial cells’ architecture and gene expression using immunohistochemistry and real-time quantitative reverse transcription (RT-qPCR).

### 2.1. Angiogenesis Is Robust in Mdx/Utrn+/− Mice

Microvessels, their constituent endothelial cells, and myofibers were visualized to assess angiogenic outcomes (Figure 1). Analysis using anti-CD31 antibody revealed a 120 ± 26.1% (*p* = 0.0206) increase of positively stained tissue in mdx/utrn+/− (Diseased) mice samples compared to 8-week-old C57BL/10ScSn (Healthy) mice (Figure 2a). Microvessel density aligned with CD31 staining results with a 78.0 ± 6.53% (*p* < 0.0001) greater in Diseased mice than Healthy mice. Further assessment of the capillary-fibre perimeter exchange (CFPE) index showed a 49.79 ± 6.9% (*p* = 0.0001) increase in Diseased samples compared to Healthy (Figure 2c). Particle analysis was used to examine microvessel architecture. Microvessels were 8.01 ± 2.84% (*p* = 0.0147) less circular in Diseased samples compared to Healthy ones (Figure 2e).

### 2.2. Endothelial Cell RNA Transcription Is Dysregulated in Mdx/Utrn+/− Mice

RT-qPCR analysis showed that endothelial-associated genes were expressed less in the Diseased cohort relative to age-matched healthy controls (Figure 3a). Surprisingly, *Cd31* expression was significantly reduced by 43.7 ± 19.4% (*p* = 0.0161). The expression levels of *Vegfr2*, *Vegfr1*, *Tie2*, and *Tie1*, genes that encode for surface receptors associated with regulating angiogenesis, were notably reduced, with fold changes ranging from approximately 50% to nearly 70% lower compared to Healthy samples. *Ang 1* and *Ang 2* expression were significantly lower in the Diseased cohort, but the proportional expression of these antagonists was unchanged compared to the Healthy cohort (Figure 4a).

### 2.3. Ang 1-Induced Alterations in the Microvascular Niche

Interestingly, gastrocnemius samples from 10-week-old mdx/utrn+/− mice that received an Ang 1 intramuscular injection (Ang 1 Treated) had 36.5 ± 13.3% (*p* = 0.0216) less CD31-positively stained tissue compared to gastrocnemius samples from that were injected with phosphate-buffered saline (PBS) (Vehicle-only) (Figure 2b). However, there was no significant difference in microvessel density (−16.0 ± 11.3%, *p* = 0.101) nor the CFPE index (−1.9 ± 10.1%, *p* = 0.7719) between Ang 1-treated and Vehicle-only samples (Figure 2d). Ang 1-treated microvessels were 10.0 ± 3.21% more circular (*p* = 0.0442) compared to Vehicle-only (Figure 2f).

Ang 1-treated gastrocnemii showed significantly greater *Vegfr2* and *Vegfr1* gene expression than Vehicle-only. *Cd31* expression increased 66.0 ± 17.4% in the Ang 1-treated samples; the *p*-value of 0.0504 suggests a trend nearing significance (Figure 3b).

### 2.4. Ang 1 Had Additional Influences on the Skeletal Muscle Niche Gene Expression

*Vegfa* gene expression was significantly increased in Ang 1-treated gastrocnemii compared to Vehicle-only (Figure 2). Notably, *Ang 1* and *Ang 2* expressions were consistent between Ang 1-treated and Vehicle-only samples, but *Ang 1* was expressed relatively more than *Ang 2* in the Ang 1-treated samples (Figure 3b).

### 2.5. DMD Mice Had Minor Changes to the Skeletal Muscle Composition

Myofiber size was consistent between Healthy and Diseased, Treated and Vehicle-only. Diseased mice had a 13.34 ± 6.87% decrease in area compared to Healthy mice (*p* = 0.0642) (Figure 5a). Ang 1-treated mice had a 15.6 ± 7.73% increase in myofiber area compared to Vehicle-only mice (*p* = 0.0792) (Figure 5b).

Diseased samples had 149.7 ± 36.3% more collagen than Healthy samples (*p* = 0.0260) (Figure 6a and Figure 7a). No significant differences were detected between Ang 1-treated and Vehicle-only samples (−0.637 ± 33.7%, *p* = 0.9847) (Figure 6b and Figure 7b).

## 3. Discussion

Investigations of functional ischemia and endothelial cell dysfunction in DMD emphasize the need to develop therapies to restore the skeletal muscle microvascular niche. Previous studies have shown Ang 1’s impact on angiogenesis. This novel study further evaluates endothelial cell alterations and angiogenic outcomes following Ang 1 delivery in the gastrocnemius microvascular niche of mdx/utrn+/− mice.

The data suggest angiogenesis was robust leading up to eight weeks of age, but was dysregulated. Muscle deterioration was the primary driving force for angiogenesis. At eight weeks of age, the mdx/utrn+/− mouse’s gastrocnemius was highly vascularized with significantly more endothelial cells and microvessels. This was expected as blood volume and flow increased at the same age in other murine DMD models [19]. The low circularity of the microvessels indicates they were new, as sprouting angiogenesis produces tortuous and disorganized microvessels that later regress [20], similar to the neovasculature of murine mammary carcinoma [21]. Despite the vascularization, however, carcinomas have leaky vessels from gaps in endothelial cells [22]. The significant increase in the CPFE index with uniform myofiber area distribution in DMD mice implies that angiogenesis was necessary to maintain sufficient gas and nutrient exchange between microvessels and myofibers [23]. This evidence questions the efficiency and effectiveness of the neovasculature in DMD mice. We speculate that endothelial cells proliferated, but chronic inflammation created a prolonged imbalance of Ang 1/Ang 2, which left the microvasculature immature and leaky, similar to carcinoma microvasculature. Paradoxically, we found that the expression of endothelial-associated genes was negatively correlated with increased microvasculature, contrary to expectations. Aiming to correct this dysfunction, we explored alterations following the delivery of exogenous Ang 1.

Results indicate Ang 1 attenuated angiogenesis, culminating in a more refined microvascular niche. Ang 1-treated myofibers show a trend towards being significantly larger, requiring more nutrients and gas exchange. Interestingly, Ang 1-treated mice achieved an average CFPE index equal to Vehicle-only mice, but with fewer endothelial cells. Ang 1-treated microvessels were less tortuous, a trait more pronounced as wounds heal and inefficient microvessels regress [20]. We believe Ang 1 reduced endothelial barrier permeability, which created a more efficient exchange of nutrients and gas. Tortuous and disorganized microvessels then began to regress as hypoxia subsided. Our lab previously found that CD31 expression was greater following Ang 1 treatment [10]. This difference may be caused by variability and requires further investigation with greater power. Considering the loss in perfusion found in other murine models of DMD, we expect a decline in CD31 expression to begin shortly after ten weeks of age.

Endothelial cells appear to have improved outcomes as the Ang 1 treatment significantly increased *Vegfr1* and *Vegfr2*, while *Cd31* strongly trended towards significance. These genes play a critical role in endothelial cell homeostasis. Ang 1 treatment did not restore expression to the same level as healthy tissues, but progress is promising. Considering the loss of endothelial cells detected, Ang 1/Tie2 signalling may be a cornerstone of proper gene expression. Microvessel functional changes should be investigated to determine the real-world impact of gene expression alterations.

The Diseased and Ang 1-treated mice’s gastrocnemius average myofiber area was comparable with Healthy and Vehicle-only controls, respectively. Diseased and Vehicle-only muscle samples tended to have fewer larger myofibers. The shift towards more small myofibers reflects the degenerative nature of Duchenne muscular dystrophy. Ang 1-treated samples tended to have larger myofibers similar to Healthy samples, suggesting Ang 1 had a knock-on effect that led to stabilization of myofiber growth. Longitudinal studies with functional tests may provide better insight into the impact of disease progression and Ang 1 treatment.

In this study, Ang 1 did not decrease collagen deposition as previously reported [10]. Differences in tissue preparation and image analysis may account for this contrasting result, highlighting the importance of optimizing the dosage, refining the delivery method, and carefully considering the treatment window. Ang 1 may have a systemic effect, confounding the outcome of Vehicle-only samples.

This project was limited in terms of time and funding availability. Profound differences between Diseased and Vehicle-only samples may be due to injection trauma, systemic effects of Ang 1, or disease progression. We could not fully process 10-week-old untreated mdx/utrn+/− samples. Furthermore, data not yet published indicates a change in RNA expression in mdx/utrn+/− mice between eight and ten weeks of age. This study aimed to further characterize mdx/utrn+/−; future studies will include age-matched untreated mdx/utrn+/− mice for more robust statistical analysis. Further studies should prioritize delivery strategies that enhance the localization of Ang 1. No sex differences were observed, but increased power may prove otherwise.

Chronic inflammation induces poor angiogenic regulation, resulting in a hostile microenvironment; the results support Ang 1’s efficacy in improving angiogenic regulation and outcomes.

## 4. Materials and Methods

### 4.1. Animals

All animal protocols were conducted in strict accordance with the Canadian Council on Animal Care and were approved by the Animal Use Subcommittee (Animal Use Protocol # 2018-140, approved in July 2018; Western University, London, ON, Canada). All experiments were performed at The Lawson Health Research Institute at St. Joseph’s Health Care (SJHC) in London, Ontario. C57BL/10ScSn and mdx/utrn+/− mice, previously validated as a model of DMD [24], were purchased from the Jackson Laboratory (Bar Harbor, ME, USA) and maintained at the Animal Care Facility at SJHC under controlled conditions (19–23 °C, 12-h light/dark cycles) and allowed water and food ad libitum. Samples were collected, analysed, and categorized into four distinct groups. Gastrocnemius samples were collected from 8-week-old C57BL/10ScSn (Healthy) and mdx/utrn+/− mice (Diseased). In addition, gastrocnemius samples were collected from the right and left hindlimbs of 10-week-old mdx/utrn+/− mice two weeks after intervention with contralateral PBS (Vehicle-only) or Ang 1 (Ang 1-Treated) injections. All groups used in this study had an *n* = 6. Each group consisted of half males and half females. Genotyping was conducted using tail snips or ear notches and polymerase chain reaction (PCR) with MyTaq™ Extract-PCR Kit (FroggaBio Inc., Concord, ON, Canada). The following set of utrophin gene primers was used: 5′-TGCAGTGTCTCCAATAAGGTATGAAC-3′, 5′-TGCCAAGTTCTAATTCCATCAGAAGCTG-3′ (forward primers) and 5′-CTGAGTCAAACAGCTTGGAAGCCTCC-3′ (reverse primer) (Sigma-Aldrich, St. Louis, MO, USA). Gel electrophoresis was used to determine the molecular weight of the amplified DNA.

### 4.2. Local Delivery of Growth Factors

The preparation and delivery of growth factors were based on previously described techniques [10]. Briefly, Affi-Gel Blue Beads (BioRad, Hercules, CA, USA) were air-dried in a cell culture hood under sterile conditions overnight. The next day, beads were resuspended in 10 μL of sterile PBS or 5 μg of recombinant human Ang 1 diluted in 10 μL of sterile PBS. Beads were incubated in the solutions overnight. The next day, beads were centrifuged for 5 min at 12,000 rpm, the supernatant was removed, and the pelleted beads were re-suspended in 10 μL of sterile PBS. Eight-week-old mdx/utrn+/− mice were aesthetically induced in a 5% oxygen-balanced isoflurane mixture. Anaesthesia was maintained with a 1.5–2% isoflurane mixture at a constant rate of 1 L/min via a nose cone. Hindlimb hair was removed using Nair, and the exposed skin was wiped with isopropyl alcohol to ensure sterility. Ang 1- or PBS-coated Affi-Gel Blue Beads were injected intramuscularly into the posterior compartments of the hindlimb (lateral head of the gastrocnemius muscle) as follows: the right hindlimb received PBS-coated beads (Vehicle-only), the left hindlimb received Ang 1-coated beads (Ang 1-Treated). Mice were recovered following the injections. Following 14 days of recovery, these mice were euthanized to collect tissue for analysis.

### 4.3. Tissue Preparation

Mice were sacrificed via carbon dioxide gas euthanasia and cervical dislocation. For histology, to ensure gastrocnemius muscles were embedded in the same orientation, the muscles were bisected across the transverse plane one-third the length of the muscle down from the superior insertion, frozen using liquid nitrogen cooled 2-methylbutane (Beantown Chemical Corporation, Hudson, NH, USA), and embedded in optimal cutting temperature compound (VWR, Radnor, PA, USA) to be sectioned from the middle to proximal and distal ends. The section slides produced were 6 μm thick. Gastrocnemius tissues used for RT-qPCR analysis were dissected and immediately flash-frozen in liquid nitrogen. All samples were stored at −80 °C.

### 4.4. Immunohistochemistry

Whole, frozen gastrocnemius samples were sectioned at Molecular Pathology at Robarts Research Institute, London, ON. Later, sections were allowed to thaw for 5 min, then fixed in 4% neutral buffered formalin (10% Formalin, 2 g H_2_PO_4_−, 3.5 g Na_2_HPO_4_ (Thermo Fisher Scientific, Waltham, MA, USA)) at room temperature for 10 min before staining. Sections were permeabilized using histology buffer (0.01% Triton X-100 (Sigma-Aldrich, St. Louis, MO, USA), 1% bovine serum albumin in PBS) for 15 min at room temperature. Background Sniper (Biocare Medical, Concord, CA, USA) was applied for 5 min to reduce nonspecific background staining. Sections were incubated with the following primary antibodies (Abcam, Cambridge, UK) at 4 °C overnight: anti-CD31 (1:10, ab28364) and anti-laminin (1:100, ab11575). All antibodies were diluted in histology buffer. Alexa fluor IgG (1:200, Life Technologies, Carlsbad, CA, USA) secondary antibodies diluted in histology buffer were used to visualize the primary antibodies. ProLong Gold anti-fade with 4′,6-diamidino-2-phenylindole (DAPI) (Life Technologies, Carlsbad, CA, USA) was added to visualize nuclei and to mount glass coverslips.

### 4.5. Microscopy and Image Analysis

Fluorescent images were acquired on an epifluorescence microscope (Nikon Eclipse Ts2R) using NIS Elements Microscope Image Software (Nikon, version 5.30.02, Minato City, Tokyo, Japan). A minimum of five non-overlapping images of three technical replicates were taken per sample at 20× magnification. ImageJ software LOCI, version 1.54f, Madison, WI, USA) was used for image analysis [25]. CD31 and laminin fluorescent signals were limited to a threshold to remove background signal noise. The percent area of positive for the signal was measured. Microvessels were identified as CD31-positive particles with a minimum area of 3.5 µm^2^. Individual myofibers stained with anti-laminin were traced using ImageJ’s freehand selection tool to define regions of interest, and their perimeter and area were measured [25]. As previously described, the CFPE index evaluated the surface area contact between microvessels and myofibers [26]. Briefly, laminin was used to trace the perimeter of myofibers. Adjoining microvessels were counted and given a share factor based on the number of adjacent myofibers. Tissue slides were sent to Molecular Pathology at Robarts Research Institute, London, ON, Canada, for Masson’s trichrome staining. Using Northern Eclipse Image software (Empix Imaging Inc., Version 8.0, Mississauga, ON, Canada), representative chromogenic images were taken on a Zeiss Axioskop Trinocular microscope (Carl Zeiss Inc., White Plains, NY, USA) at 20× magnification. Five non-overlapping images of three technical replicates were taken per sample. Masson’s trichrome staining was used quantitatively to assess collagen deposition using ImageJ software [25]. Fibrosis is excessive collagen deposition in muscles.

### 4.6. RNA Extraction and cDNA Preparation

Frozen gastrocnemius tissue was cut and weighed, producing 40–60 mg of tissue. Samples were subjected to three freeze–thaw cycles using liquid nitrogen, homogenized using polypropylene pestles (Fisher Scientific, Hampton, NH, USA), and then incubated in Trizol Reagent (Invitrogen, Waltham, MA, USA) for a minimum of 10 min on ice. RNA purification was performed using a Direct-zol RNA miniprep kit (Zymo Research, Orange, CA, USA) per the manufacturer’s instructions. An in-solution DNase I treatment was performed post-purification using <10 µg of RNA sample, DNase I (Zymo Research, Orange, CA, USA), DNA Digestion Buffer (Zymo Research, Orange, CA, USA), and water incubated at room temperature for 15 min. Three volumes of Trizol were added to one volume of the sample solution. RNA purification using a Direct-zol RNA miniprep kit was repeated and suspended in 40 µL of RNAase-free water. RNA concentration and quality were quantified using a DS-11 spectrophotometer (DeNovix, Wilmington, DE, USA). All samples were verified to have a 260/280 ratio above 1.9 and a 260/230 ratio between 2.0–2.2.

cDNA was produced using 1.5 μg of total RNA using the High-Capacity cDNA Reverse Transcription Kit (Applied Biosystems, Waltham, MA, USA) per the manufacturer’s instructions.

### 4.7. RT-qPCR

TaqMan Gene Expression Assays (Thermofisher, Waltham, MA, USA) was used to assay *Cd31* (*Pecam1* Mm01242576_m1), *eNos* (*Nos3* Mm00435217_m1), *Vegfr2* (*Kdr* Mm01222421_m1), *Vegfr1* (*Flt1* Mm00438980_m1), *Vegf* (*Vegfa* Mm00437306_m1), *Ang 1* (*Angpt1* Mm00456503_m1), *Ang 2* (*Angpt2* Mm00545822_m1), *Tie1* (*Tie1* Mm00441786_m1) *Tie2* (*Tek* Mm00443243_m1), *Csnk2a2* (*Csnk2a2* Mm01243455_m1), *Ap3d1* (*Ap3d1* Mm00475961_m1), *Actb* (*Actb* Mm02619580_g1). RT-qPCR samples were produced and measured on the QuantStudioTM 5 system (Thermo Fisher Scientific, Waltham, MA, USA) with TaqMan Fast Advanced Master Mix (Applied Biosystems, Waltham, MA, USA). Quantification was performed using Design & Analysis Software 2.6 (Thermo Fisher Scientific, Waltham, MA, USA). Total *Cd31*, *eNos*, *Vegfr1*, *Vegfr2*, *Vegf*, *Ang 1*, *Ang 2*, *Tie1*, and *Tie2* expression were normalized to the geometric mean of established control genes: *Csnk2a2*, *Ap3d1*, and *Actb* [27].

### 4.8. Statistical Analysis

To properly analyse data sets, Shapiro–Wilk tests for parametric distribution, F-tests for variance in parametric data, and appropriate significance tests were performed using GraphPad Prism for Windows (GraphPad Software, version 8.0.0, San Diego, CA USA). Data collected were determined to be parametrically distributed with a *p*-value > 0.05 and have equal variance with a *p*-value > 0.05. Data derived from Healthy and Diseased gastrocnemius samples were tested for significance using unpaired t-tests when data sets were parametric. Mann–Whitney U tests were used for nonparametric data. Welch’s correction was used when parametric data sets had unequal variance. Vehicle-only and Ang- 1-treated gastrocnemius samples were compared using paired t-tests for parametrically distributed data and the Wilcoxon matched-pairs signed rank test for nonparametrically distributed data.

## Figures and Tables

**Figure 1 ijms-25-11824-f001:**
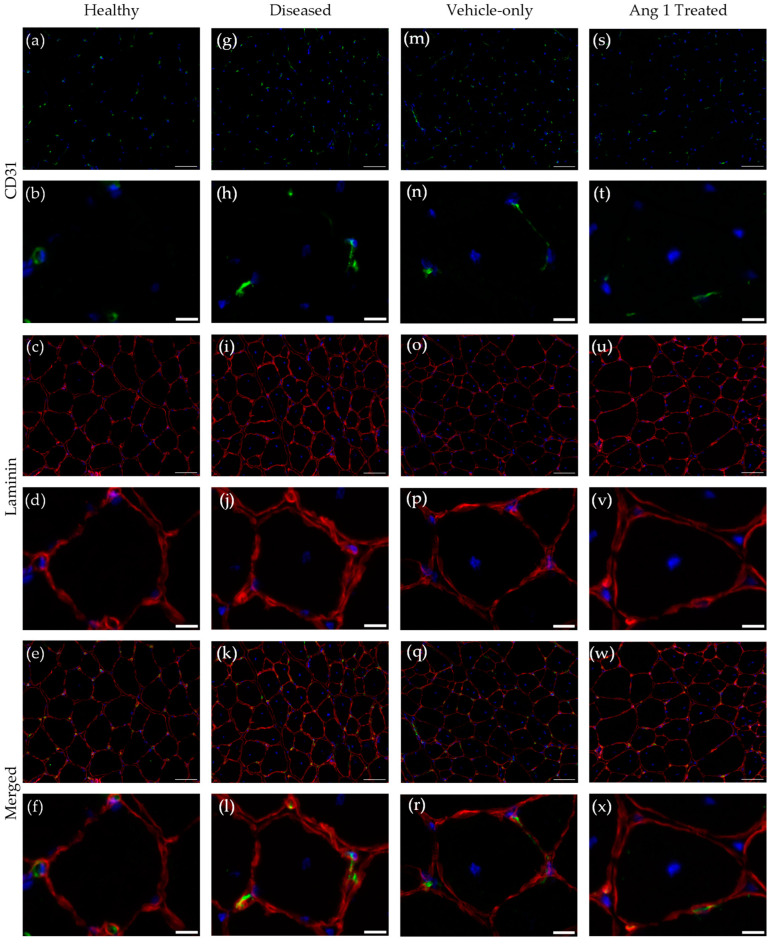
Histology of anti-CD31-stained endothelial cells and anti-laminin-stained myofibers in Healthy (**a**–**f**), Diseased (**g**–**l**), Vehicle-only (**m**–**r**), and Ang 1-Treated (**s**–**x**) transverse gastrocnemius samples. Scale bar = 50 µm (**a**,**c**,**e**,**g**,**i**,**k**,**m**,**o**,**q**,**s**,**u**,**w**), 10 µm (**b**,**d**,**f**,**h**,**j**,**l**,**n**,**p**,**r**,**t**,**v**,**x**).

**Figure 2 ijms-25-11824-f002:**
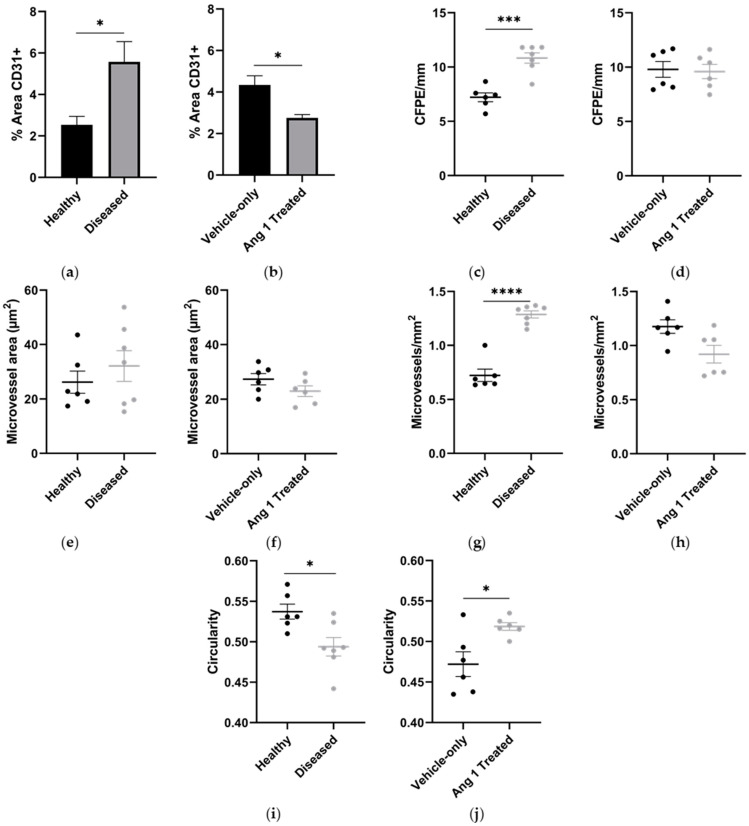
Microvessel morphometry in Healthy, Diseased, Vehicle-only, and Ang 1-Treated gastrocnemius samples: (**a**,**b**) percent area CD31 positively stained tissue, (**c**,**d**) capillary-to-fibre perimeter exchange (CFPE) index, (**e**,**f**) microvessel area, (**g**,**h**) microvessel density, and (**i**,**j**) microvessel circularity. Data are mean ± SEM. *n* = 6–7. * *p* < 0.05, *** *p* < 0.001, **** *p* < 0.0001.

**Figure 3 ijms-25-11824-f003:**
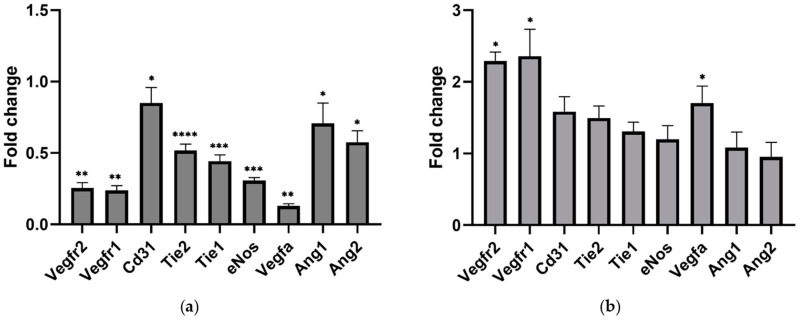
Analysis of endothelial cell-related gene expression by Reverse Transcription Quantitative Polymerase Chain Reaction (RT-qPCR). Gene expression was measured in the gastrocnemius of (**a**) Diseased relative to Healthy and (**b**) Ang 1-Treated relative to Vehicle-only. Data are mean ± SEM. *n* = 6. * *p* < 0.05, ** *p* < 0.01, *** *p* < 0.001, **** *p* < 0.0001.

**Figure 4 ijms-25-11824-f004:**
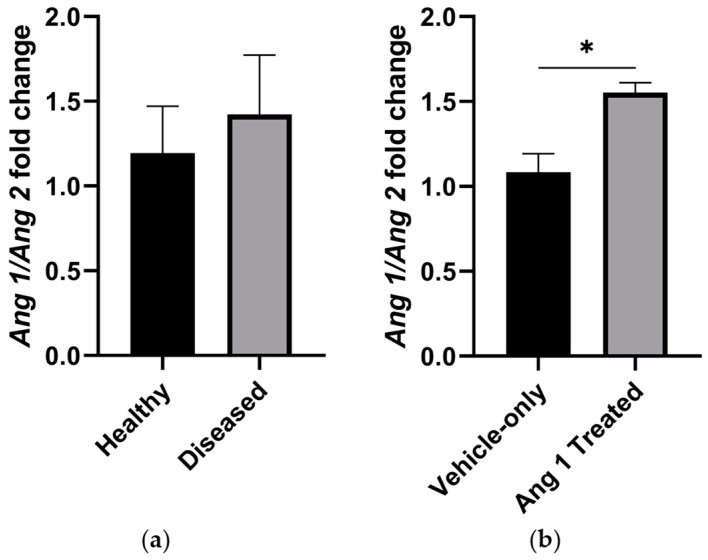
*Ang 1*:*Ang 2* relative fold change ratio of (**a**) Healthy and Diseased and (**b**) Vehicle-only and Ang 1-Treated gastrocnemii. Data are mean ± SEM. *n* = 6. * *p* < 0.05.

**Figure 5 ijms-25-11824-f005:**
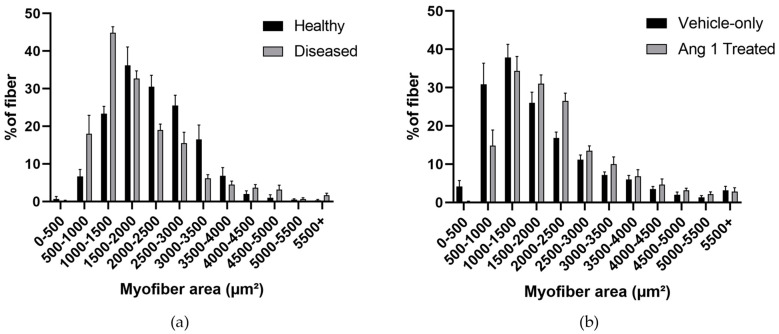
Distribution of myofiber area of (**a**) Healthy, Diseased *p* = 0.0642, (**b**) Vehicle-only, and Ang 1-Treated *p* = 0.0792 gastrocnemius samples. *n* = 6.

**Figure 6 ijms-25-11824-f006:**
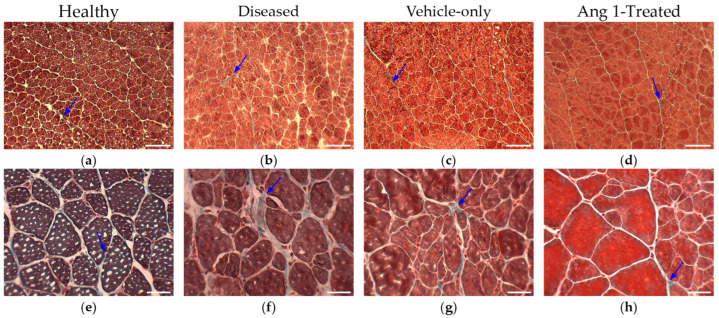
Representative histology of Masson’s Trichrome staining. Blue arrows indicate collagen deposition in (**a**) Healthy, (**b**) Diseased, (**c**) Vehicle-only, and (**d**) Ang 1-Treated gastrocnemius with scale bar = 150 μm. (**e**) Healthy, (**f**) Diseased, (**g**) Vehicle-only, and (**h**) Ang 1-Treated gastrocnemius with scale bar = 35 μm.

**Figure 7 ijms-25-11824-f007:**
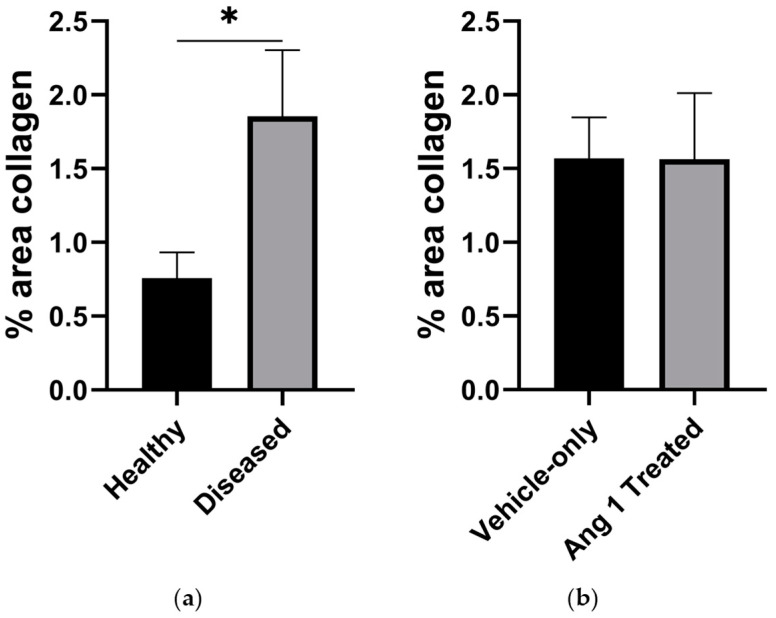
Quantitative analysis of collagen deposition in (**a**) Healthy and Diseased, and (**b**) Vehicle-only, and Ang 1-Treated gastrocnemius. *n* = 6. * *p* < 0.05.

## Data Availability

Please contact the corresponding author with any data inquiries.

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
