# Peer review of "Angiopoietin 1 Attenuates Dysregulated Angiogenesis in the Gastrocnemius of DMD Mice"

_ijms, 2024, doi:10.3390/ijms252111824_

Round 1
Reviewer 1 Report (Previous Reviewer 1)
Comments and Suggestions for Authors
In the current study, the authors tested the role of angiopoietin 1 on the skeletal muscle microvasculature using a mouse DMD model. This study has a lot of deficiencies in the experimental design, that make the interpretation of the results difficult. Re-writing and re-submitting the manuscript have not eliminated these deficiencies. This diminishes the significance and the conclusions of this study.
Suggestions:
Line 20: “Gene expression in Ang gastrocnemii increased, while myofiber size distribution was consistent”. This is vague and unclear. Which genes? Consistent with what? Please revise.
Please provide an explanation at the beginning of Results as to why you have used 8-week-old mdx/utrn+/- (Diseased) mice compared to 8-week-old C57BL/10ScSn (Healthy) mice and used 10-week-old mdx/utrn+/- (Vehicle-only) mice and 10-week-old mdx/utrn+/- (Ang1 Treated) mice. Therefore, you had to use a T-test instead of ANOVA and could not directly compare Diseased and Vehicle-only samples. Explain why you could not sacrifice all groups at the age of 10 weeks.
Muscle fibers in Figure 1 Vehicle-only group look much larger and more uniform in size than in the Diseased group. Does it mean that the vehicle has a positive effect on muscle regeneration/structure? Perhaps this is the age difference effect? Or maybe the systemic Ang1 circulation due to the method used since samples were collected from the right and left gastrocnemii of the same 10-week-old mdx/utrn+/- mice following contralateral PBS (Vehicle-only) or Ang 1 (Ang 1 Treated) injections.
Line 135: “Myofibers size was consistent between both groups”. What does this mean? Please revise.
Figure 6F shows that muscle fibers in the Diseased group are larger than in the Healthy group (6E). This makes no sense since Figure 5a shows the opposite distribution of fiber sizes.
Line 205: “The Diseased and Ang 1 Treated mice's gastrocnemius myofiber area was comparable with Healthy and Vehicle-only controls, respectively”. The average size was similar, but Figure 5 shows clear differences that need to be discussed.
Line 214: “the results support Ang 1’s efficacy in improving angiogenic regulation and outcomes”. There is a very subtle difference between Vehicle-only and Ang 1 Treated groups. The main reason is probably the systemic effect of injected Ang1 due to the suboptimal method used in this study.
There are a lot of limitations in this study, and they are not clearly listed in the manuscript. There should be limitations of the study section at the end of the Discussion.
Author Response
Line 20: “Gene expression in Ang gastrocnemii increased, while myofiber size distribution was consistent”. This is vague and unclear. Which genes? Consistent with what? Please revise.
This has been revised in the manuscript.
Please provide an explanation at the beginning of Results as to why you have used 8-week-old mdx/utrn+/- (Diseased) mice compared to 8-week-old C57BL/10ScSn (Healthy) mice and used 10-week-old mdx/utrn+/- (Vehicle-only) mice and 10-week-old mdx/utrn+/- (Ang1 Treated) mice. Therefore, you had to use a T-test instead of ANOVA and could not directly compare Diseased and Vehicle-only samples. Explain why you could not sacrifice all groups at the age of 10 weeks.
At the beginning of the results section, we stated that this project's scope was initially to characterize the baseline further. We have included the explanation as part of the limitations of this study and included recommendations.
Muscle fibers in Figure 1 Vehicle-only group look much larger and more uniform in size than in the Diseased group. Does it mean that the vehicle has a positive effect on muscle regeneration/structure? Perhaps this is the age difference effect? Or maybe the systemic Ang1 circulation due to the method used since samples were collected from the right and left gastrocnemii of the same 10-week-old mdx/utrn+/- mice following contralateral PBS (Vehicle-only) or Ang 1 (Ang 1 Treated) injections.
Due to a clerical error, the images used in Figure 1 m-r (Vehicle-only) were from a Diseased sample. Representative Vehicle-only images are presented correctly in the revised version. In the discussion section, we further discussed the variability of myofibers.
Line 135: “Myofibers size was consistent between both groups”. What does this mean? Please revise.
This has been revised to provide more clarity.
Figure 6F shows that muscle fibers in the Diseased group are larger than in the Healthy group (6E). This makes no sense since Figure 5a shows the opposite distribution of fiber sizes.
The image was initially chosen because it showcased collagen deposition well. However, we changed the sample image for Figure 6f to another image more representative of myofiber size while showing collagen deposition.
Line 205: “The Diseased and Ang 1 Treated mice's gastrocnemius myofiber area was comparable with Healthy and Vehicle-only controls, respectively”. The average size was similar, but Figure 5 shows clear differences that need to be discussed.
We have added a discussion of the trending change in myofiber size distribution.
Line 214: “the results support Ang 1’s efficacy in improving angiogenic regulation and outcomes”. There is a very subtle difference between Vehicle-only and Ang 1 Treated groups. The main reason is probably the systemic effect of injected Ang1 due to the suboptimal method used in this study.
We revised the discussion to include more on the systemic effect of injected Ang 1.
There are a lot of limitations in this study, and they are not clearly listed in the manuscript. There should be limitations of the study section at the end of the Discussion.
In the discussion, we have further clarified the limitations of this study.

Reviewer 2 Report (New Reviewer)
Comments and Suggestions for Authors
The authors aimed to elucidate changes in the DMD mouse’s gastrocnemius microvascular niche following local administration of Ang 1.
The scientific facts are acceptable.
Does 10-week-old 228 mdx/utrn+/- mice exhibit exhibit a muscular dystrophy similar to what the authors require? Please cite a reference if possible.
Can the method adapted by the authors for sacrifice of animals contribute to the additional stress and influence the findings?
What is the effect of Ang 1 on the animals with regard to different weeks of treatment?
Ang 1 stimulates processing of pro-HB-EGF by metalloproteinases, and the released HB-EGF transactivates EGFR to induce angiogenesis. Please discuss this fact.
Perhaps, the authors could add the facts- two of the Angs, Ang-1 and Ang-4, activate the Tie2 receptor, whereas Ang-2 and Ang-3 inhibit Ang-1-induced Tie2 phosphorylation.
Few limitations could be added.
Author Response
Does 10-week-old 228 mdx/utrn+/- mice exhibit exhibit a muscular dystrophy similar to what the authors require? Please cite a reference if possible.
This model was previously validated, and a reference has been included.
Can the method adapted by the authors for sacrifice of animals contribute to the additional stress and influence the findings?
We are unaware of any confounding effects of CO2 gas euthanasia and cervical dislocation on our findings.
What is the effect of Ang 1 on the animals with regard to different weeks of treatment?
We have yet to investigate different courses of treatment, but this is a question we intend to answer as soon as possible.
Ang 1 stimulates processing of pro-HB-EGF by metalloproteinases, and the released HB-EGF transactivates EGFR to induce angiogenesis. Please discuss this fact.
The article discussing this concept has been retracted due to falsified data. Discussing it may introduce misleading information into the manuscript.
Perhaps, the authors could add the facts- two of the Angs, Ang-1 and Ang-4, activate the Tie2 receptor, whereas Ang-2 and Ang-3 inhibit Ang-1-induced Tie2 phosphorylation.
We have acknowledged the roles of Ang 3 and 4 in the introduction.
Few limitations could be added.
In the discussion, we have further clarified the limitations of this study.

Round 2
Reviewer 1 Report (Previous Reviewer 1)
Comments and Suggestions for Authors
The authors addressed some of the reviewer’s critiques. The manuscript was improved after these modifications. Although I still think that this study has a lot of deficiencies in the experimental design, I think it might be of some interest for the readers to have access to the results of this investigation.
Suggestions:
I suggest moving the text from Line 72 to Line 61.
Figure 6.” Magenta arrows indicate collagen deposition”. There are no visible arrows. Bars are practically invisible in this Figure.
Author Response
Thank you for your valuable input throughout the revision process. Your insight and recommendations have greatly improved the quality of our manuscript.
I suggest moving the text from Line 72 to Line 61.
We moved Line 72 to Line 61.
Figure 6.” Magenta arrows indicate collagen deposition”. There are no visible arrows. Bars are practically invisible in this Figure.
We added blue arrows to the images and updated the caption accordingly. We enlarged the scale bars and colored them white to enhance clarity.
This manuscript is a resubmission of an earlier submission. The following is a list of the peer review reports and author responses from that submission.
Round 1
Reviewer 1 Report
Comments and Suggestions for Authors
In the current study, the authors test the role of angiopoietin 1 on the skeletal muscle microvasculature using a mouse DMD motel. The topic is interesting, but the study has a lot of deficiencies, including the experimental design, that make the interpretation of results difficult and confusing. These deficiencies need to be addressed before making any conclusions about the study results.
Suggestions:
Line 20: “These results suggest robust angiogenesis in DMD mice, but endothelial cells lack essential gene expression—furthermore, exogenous Ang 1 attenuated angiogenesis. Consequentially, myofibers grew (you did not measure this; fiber diameters were increased but was it from better regeneration or fewer degenerated fibers?), and gene expression increased”. This is vague and unclear. Please revise.
Line 30: “resulting in skeletal muscle wasting and impairment of the microvascular niche”. This required further explanations and references. Due to the significant DMD truncation, muscle fibers experience cycles of degeneration and regeneration via activation of the satellite cells and new fiber formation. Moreover, there is a significant fibrosis and substitution of muscle fibers with the connective tissue that is less vascularized than skeletal muscle.
Line 74: “Ang 1 is linked to muscle regeneration through myogenin induction, which promotes myogenesis”. This required further explanations. Ang-1 enhances muscle fiber regeneration in cardiotoxin-injured TA muscle via enhanced satellite cell activation and differentiation and this is also shown in culture and is independent of the vasculature (18). This suggests that Ang 1 can act on both (and probably independently): improved vasculature and enhanced satellite cell activation and differentiation.
Figure 2. Please modify this figure by combining Healthy//Diseased/Vehicle only/Treated for % Area, CFP, Circularity, and Microvascular area and quantify significant differences using One-Way ANOVA. You analyzed four different conditions in your experience. It is not clear to me why you separated them into two different groups for this and other figures.
Figures 3 and 4. Similar to the previous figure, please combine a and b into one figure, normalize the expression to a healthy control, and quantify significant differences using One-way ANOVA.
Line 97: “aged-matched”. Age-matched?
Line 136: “Myofibers of Diseased mice had a 7.07 ± 3.29% decrease in perimeter compared to Healthy”. Changes in muscle fibers are usually evaluated by comparing the muscle fiber area, not the perimeter. Please recalculate. The most informative are diagrams showing distributions in different fiber sizes. Figure 8d clearly shows the presence of a large number of small, regenerated muscle fibers similar to Figure 8b. The diagram will show whether there are some changes in the size of regenerated fibers between these samples.
Figure 5. Similar to the previous figures, please combine a and b into one figure, normalize the fiber area of healthy control, and quantify significant differences using One-way ANOVA.
Figure 6. It is not clear whether these are positive myogenic cells or positive myonuclei within muscle fibers. Please use laminin staining to visualize muscle fibers to evaluate this.
Figure 7. Same suggestion as for Figure 5.
Why vehicle vehicle-only treatment reduces % of MYOG-positive nuclei when compared with the diseased samples? Is this due to the whole-body circulation of the Ang 1 from the contralateral hindlimb?
Figure 8. Please show low magnification in addition to the high magnification presented in this figure. Usually, there is a large difference in the amount of connective tissue in the endomysium shown in Figure 8a and perimysium shown in Figure 8b and Figure 8d. It is unclear what was quantified and compared: endomysium, perimysium, or a mix of both (which makes no sense).
Figures 6 and 7. Please add QRT-PCR evaluations of MYOG in these samples. It should be easy to do since you already have mRNA samples. This is a much more sensitive and quantitative technique that will add clarity to this question.
Figure 9. Similar suggestions as to the previous figures. One figure should be made and One-way ANOVA should be used.
Line 171: “highly vascularized with significantly more endothelial cells and microvessels”. This is expected since there are many more small-diameter muscle fibers each of which is surrounded by capillaries.
Line 176: “Despite the vascularization, however, tumours have hypoxic regions and leaky vessels”. Which tumors?
Line 181: “chronic inflammation created a prolonged imbalance of Ang 1/Ang 2, which left the microvasculature immature and leaky”. This was not measured in this study.
Line 184: “Endothelial ineffectiveness”. It was not measured in this study. What justifies this statement?
Line 191: “We believe Ang 1 reduced endothelial barrier permeability, which created a more efficient exchange of nutrients and gas”. It was not measured in this study. What justifies this statement?
Line 191: “Results indicate Ang 1 attenuated angiogenesis, culminating in a more refined microvascular niche”. If Ang 1 improved regeneration, then there are fewer small-diameter muscle fibers, and therefore fewer associated capillaries.
Line 208: “The lack of functional dystrophin likely obviated Ang 1’s reported abilities to induce myogenesis, so we speculate that Ang 1 induced a suitable microenvironment through endothelial cell maturation, which allowed for better myofiber survival.” This statement is not justified. This study has not measured myofiber survival. Lack of dystrophin does not affect the ability of satellite calls to be activated and undergo differentiation forming new myofibers. Ang-1 enhances muscle fiber regeneration in cardiotoxin-injured TA muscle via enhanced satellite cell activation and differentiation and this is also shown in culture and is independent of the vasculature (reference18).
Line 208: “In this study, Ang 1 did not decrease collagen deposition”. This statement is not justified since it is unclear which collagen deposits were measured and compared: endomysial or perimysial.
Line 223: Please add Animal Use Protocol approval date.
Line 232: “10-week-old mdx/utrn+/- mice following contralateral PBS (Vehicle-Only) or Ang 1 (Treated) injections”. Ang1 is soluble after it is detached from beads. There is no discussion in the study on how soluble glycoprotein could affect the contralateral hindlimb.
Line 234: “Each group consisted of half males and half females”. Noting is mentioned in the results about the effect of gender on the results. This has to be discussed and clarified. Were there sex differences in response to treatment?
Line 250: “aesthetically induced in a 5% oxygen-balanced 250 isoflurane mixture”?
Line 280: Please provide the catalog numbers and the dilutions of all antibodies.
Line 280: “Adjoining microvessels were counted and given a share factor based on the number of adjacent myofibers. Ten myofibers”?
Comments on the Quality of English LanguageMinor editing of English language required. Please see comments.
Reviewer 2 Report
Comments and Suggestions for Authors
I have following concerns:
The title needs to be clear on what changes after Ang1 treatment.
All Figures have to clarify the term “treated“ by what?
Figure 1, 4, 8: the quality of histology and immunohistochemistry is low. You need to enhance background for easier comparison. Most important is that you need to provide low magnification with high magnification to convince readers.
Fig.2a and 2b are not consistent with Fig.1 images.
Fig.3, 4, qPCR results: What’s your internal control? Did you normalize your data?
Fig.6 and Fig.7 can be combined. Same as Fig.8 and Fig.9
Fig.9b has no change, which is different from your prior report. Can you explain the difference.